# Case Report: Spontaneous Left Inferior Epigastric Artery Injury in a COVID-19 Female Patient Undergoing Anticoagulation Therapy

**DOI:** 10.3390/jcm12051842

**Published:** 2023-02-25

**Authors:** Hristo Abrashev, Julian Ananiev, Ekaterina Georgieva

**Affiliations:** 1Department of Special Surgery, Orthopedics, and Traumatology, Clinic of Vascular Surgery, Medical Faculty, Trakia University, 6000 Stara Zagora, Bulgaria; 2Department of General and Clinical Pathology, Forensic Medicine and Deontology and Dermatovenereology, Medical Faculty, Trakia University, 6000 Stara Zagora, Bulgaria; 3Department of Chemistry and Biochemistry, Medical Faculty, Trakia University, 6000 Stara Zagora, Bulgaria

**Keywords:** COVID-19, SARS-CoV-2 infection, massive retroperitoneal hematoma, anticoagulant therapy, spontaneous retroperitoneal bleeding

## Abstract

Since the beginning of the pandemic, a recommendation was made for the use of anticoagulants in high-risk hospitalized patients. This therapeutic approach has positive and negative effects regarding the outcome of the disease. Anticoagulant therapy prevents thromboembolic events, but it can also lead to spontaneous hematoma formation, or be accompanied by massive active bleeding. We present a 63-year-old COVID-19-positive female patient with a massive retroperitoneal hematoma and spontaneous left inferior epigastric artery injury.

## 1. Introduction

COVID-19 usually presents as acute respiratory distress syndrome (ARDS) and affects lung tissue, although extrapulmonary involvement of several organs and systems can occur [1]. The most vulnerable groups consist of elderly patients and those with multiple comorbidities, such as chronic respiratory and preexistent cardiac disease, immunocompromised state, cancer, etc. [2,3]. Microthrombotic events are increasingly described as a hallmark of COVID-19 and are associated with complications such as pulmonary thromboembolism and respiratory failure. The effector mechanisms of anticoagulant prophylaxis are well studied and it has been found that anticoagulant therapy can significantly reduce the incidence of these conditions and mortality. On the other hand, the therapeutic dosage of anticoagulants can lead to a serious number of concerning complications such as an increased risk of bleeding [4]. Spontaneous retroperitoneal hematoma (SRH) is considered a rare complication which is characterized as spontaneous bleeding in the retroperitoneal space and is not generated by traumatic injury. It is usually observed in patients with coagulation disorders, malignancies, hematologic diseases, and elderly patients or those undergoing anticoagulation therapy or hemodialysis [5].

This case report aims to mark the potential complications and life-threatening conditions which may occur in COVID-19 patients undergoing standard LMWH therapy.

## 2. Case Presentation

A 63-year-old Caucasian woman presented to our hospital with a complaint of intermittent shortness of breath for two weeks and a dry nonproductive cough for one week, headache, fever, and fatigue. The patient was admitted to the pulmonology department with a positive PCR test with a typical X-ray finding of bilateral COVID-19 pneumonia with extensive bilateral pulmonary infiltrates involving both lower lobes. Radiographs show interstitial infiltrative inflammatory changes, mostly subpleural with a tendency to confluence. The changes are manifested in the appearance of reticulonodular changes in bilateral areas of consolidation of the parenchyma, located apical and basal (Figure 1). Admission observations were as follows: oxygen saturation of 91% on room air, blood pressure of 110/60 mmHg, heart rate of 110 beats per minute, a temperature of 38.4 °C, and pale in color. Physical examination revealed a respiratory rate of 27 bpm and bibasal wet rales during auscultation. The patient had a remarkable medical history with a serious number of associated diseases, including diabetes (type 2), and also had chronic kidney and heart failure, and arterial hypertension for which they were regularly monitored. Before hospitalization, the patient had no evidence of an acute vascular failure requiring urgent reconstructive intervention.

### 2.1. Laboratory Test Results

The patient’s laboratory results and X-ray imaging met the criteria for the severe clinical presentation of COVID-19 infection. Blood work on admission showed leukocytosis (WBC 11.6 g/L), anemia (HGB 100 g/L), chronic kidney failure with low glomerular filtration rate (19.0 mL/min/1.73 m^2^), elevated C-reactive protein level of 58.5 mg/L, Lactate dehydrogenase (LDH) of 654 U/L, Glucose 19.6 mmol/L, and D-dimer 3.6 mg/L. Arterial blood gas analysis revealed respiratory alkalosis and hypoxemia with a pH of 7.49 (normal 7.35–7.45), a CO_2_ pressure of 32 mmHg (normal 35–40), and an oxygen pressure of 37.9 mmHg (normal 80–100) (Table 1).

### 2.2. Therapy Applied

The patient’s therapy included intravenous ceftriaxone 1.0 g (twice a day), dexamethasone (6 mg daily), furosemide (40 mg twice a day), Fraxiparine 0.6 mL s.c (twice a day), insulin act rapid according by the recommended scheme, infusion of vitamins and high-flow nasal oxygen therapy. On the 7th day, the patient’s condition worsened, which was confirmed by an increased oxygen demand (the oxygen need from 6 L/h went up to 10 L/h), X-ray findings, and laboratory results, and despite the treatment, they were transferred to the intensive care unit. Ten days after hospitalization, the patient became anuric with high creatinine and urea levels, therefore hemodialysis was initiated. 

### 2.3. CT Scan Report

Between the 7th and 10th day, the patient’s HGB levels gradually dropped from 91 g/L (on day 7) to 61 g/L (on day 10) (see Table 1) with no overt evidence of upper and lower gastrointestinal tract bleeding but subsequent blood replacement was commenced. The abdominal exam showed tenderness in the lower quadrants of the abdomen that radiated to the lower back but no focus or peritoneal guarding. A routine abdominal ultrasound also revealed no evidence of intraperitoneal/retroperitoneal bleeding or any other abdominal organ disorder. Immediately after the dialysis, the patient reported acute, unbearable abdominal pain irradiating to the left flank. A computed tomography (CT) of the thorax and abdomen was performed to clarify the present abdominal drama. The prolonged clinical manifestation was unequivocally confirmed by the CT scan report. It demonstrated a left rectus sheath hematoma (RSH) and non-traumatic spontaneous retroperitoneal bleeding (Figure 2). The bleeding is continuous, and the hematoma is dissected down to the pelvis. Contrast-enhanced CT revealed two hyperdense lesions with a relatively homogeneous texture (radio density of 12 Hounsfield units (HU)) in the lower abdominal segments and the small pelvis with approximate measurements up to 25 cm on the left flank with evidence of imbibition of the left psoas muscle, internal oblique muscle, rectus abdominis muscle, and thickened anterior abdominal wall on the left side (Figure 2 and Figure 3).

A decision was made to perform an urgent midline exploratory laparotomy to identify the source of bleeding and to evacuate the retroperitoneal hematoma. Intraoperatively, a spontaneous tear was found on the left inferior epigastric artery which was sutured, and the retroperitoneal hematoma was evacuated with retroperitoneal drainage (Figure 4).

Postoperatively, the patient required hemodynamic support with vasopressors for the following 48 h, various transfusions with red blood cells, platelets, and fresh frozen plasma. The patient deteriorated and, despite adequate treatment, died on the third day after the operation.

## 3. Discussion

The high mortality in COVID-19 patients is due to various complications, among which are venous thromboembolism and disseminated intravascular coagulation as a consequence of the systemic inflammatory response and immunologically modulated reactions [6]. A multicenter study showed an overall incidence of thrombotic complications of 9.5% and an overall bleeding rate of 4.8% (with 5.65% in critically ill patients) [7]. The infection can lead to pathogen-induced immunologically mediated vasculitis expressed as an acute inflammatory reaction with significant endothelial dysfunction inducing blood stasis and thrombotic events [8]. Patients with severe clinical presentation are at high risk of systemic coagulopathy which mostly appears as thromboembolic events rather than bleeding tendencies [5]. It assumed that the mechanism of occurrence of COVID-19-associated thrombosis is due to the attachment of the SARS-CoV-2 virus to the angiotensin-2 receptor of the endothelial cells with its spike protein, leading to the release of proinflammatory cytokines (internal injury inside the vascular wall) and subsequent systemic inflammation [9]. Elevated D-dimer, prolonged PT, and thrombocytopenia have been observed in hospitalized COVID-19 patients with the frequency of these hematological changes being between 20 and 50%. These patients are characterized by a high risk for more thrombotic than hemorrhagic events [10]. D-dimer levels higher than 1.5 μg/mL may increase the incidence of venous thrombosis and a negative predictive value (94.7%) in severe COVID-19, accompanied by pneumonia. Hospitalized patients should receive a prophylactic dose of LMWH, except for those with active bleeding and a platelet count <25 × 10^9^/L [11]. D-dimer levels in the patient we describe increased during the course of the disease despite the administered therapeutic dose of anticoagulant. Simultaneously, the PTL levels showed an increased blood viscosity of 256 × 10^9^/L (reference range 130–140 × 10^9^/L) on the first day of hospitalization (Table 1). Based on the general worsening condition and multiple risk factors (pneumonia, elevated D-dimer 28.4% ≥2 times ULN, chronic diseases, and age > 60) [12], the patient was assessed as at risk for thromboembolic events. This necessitated the introduction of a therapeutic anticoagulant regimen according to recommendations for treating patients with moderate-to-severe coronavirus infection. Hypercoagulation has been the subject of great interest and extensive research, while hemorrhagic events remain poorly studied and debated in COVID-19 cases. Recently, the number of studies reporting spontaneous bleeding in under-anticoagulation-therapy COVID-19 patients has significantly increased [13]. The risk of spontaneous bleeding is increased by several factors: imbalance in platelet production and disruption, excessive consumption of coagulation factors, severe cough, mechanical ventilation, thrombocytopenia, hypertension, etc. Anticoagulation therapy in COVID-19 patients is classified by its intensity as follows: standard, intermediate, and therapeutic intensity [14]. There is considerable controversy among studies regarding the benefits and harms of using anticoagulants in hospitalized COVID-19 patients. There also continues to be considerable debate regarding the type, dose, and duration of anticoagulant medication in specific patients [15]. Our patient started on an anticoagulant therapeutic dosage and changed postoperatively to unfractionated heparin. Low-molecular-weight heparin (LMWH) is associated with low mortality, improved markers of cell death, and curtailed viral persistence with significant beneficial effects on hemostasis and a high safety profile [16]. In addition, unfractionated heparin has the ability to competitively bind to spike proteins and to inhibit viral entry, thus reducing virus infectivity [17]. The positive effect of the administration of anticoagulants can also be seen when comparing in-hospital mortality in ICU patients on anticoagulants and non-anticoagulant receivers. The group of patients receiving anticoagulants live 7 days longer [18]. Other studies report that anticoagulant therapy does not reduce the incidence of thromboembolic events and has no beneficial impact on the outcome of the disease, but anticoagulant therapy could lead to additional unanticipated complications such as hemorrhagic events [19].

The RSH risk factors Include anticoagulation, concomitant therapy with fibrinolytic agents or glycoprotein inhibitors, increasing number of comorbidities, hematological disorders, surgical interventions, trauma, rectus muscle contractions, etc. [20,21]. It is believed that the mortality rate in the general population is around 4%, while in patients of anticoagulant therapy may increase to 25%. According to the literature, RSH is 2–3 times more frequently observed in women than men, and the mean patient age varies from 46 to 69 years [22]. As a result of viral infections, disseminated intravascular coagulation (DIC) and hemorrhage may occur, represented as gastrointestinal bleeding, intracranial hemorrhage, hemoptysis, skin mucosa, internal bleeding, pulmonary/renal hemorrhage, etc. [23]. The most common localization of provoked spontaneous and severe hematomas in COVID-19 patients is seen in the iliopsoas, vastus intermedius, gluteus, sternocleidomastoid, pectoralis major muscles, and muscles of the anterior abdominal wall [24]. Jiménez et al. reported that two-thirds of hemorrhagic events are seen in patients on therapeutic anticoagulation. The highest pooled incidence estimate of spontaneous bleeding was reported for patients receiving intermediate or therapeutic dose anticoagulation (21.4%). Usually, RSH is a self-limited hemorrhage and responds well to conservative treatment after cessation of anticoagulation therapy [25]. On the tenth day of hospitalization, our patient developed a left rectus sheath hematoma (RSH), which evolved into a massive retroperitoneal hematoma. In the present case, a median exploratory laparotomy was performed due to the clinical progression of the bleeding. Intraoperatively, a partial lesion of the left inferior epigastric artery was found (Figure 4). Unfortunately, the left rectus sheath’s hematoma progressed into a massive retroperitoneal hematoma without any pronounced clinical presentation but with a considered drop in hemoglobin levels (Table 1). The incidence of spontaneous retroperitoneal hematoma in COVID-19 patients remains unclear, but studies report relatively higher rates (3–4 times higher) in patients under anticoagulant therapy [26]. The SRH etiopathogenesis could be of either parenchymal or vascular origin. The mortality rate in the current literature is about 26.8% with an incidence ranging from 0.1% to 0.6% [27]. Diagnosis can be difficult while clinical signs are often vague and nonspecific. Multiphasic CT scan with contrast remains the gold standard for the invasive diagnostics of a spontaneous retroperitoneal hematoma (Figure 2A,B). It allows active bleeding to be recognized early and the bleeding vessel to be easily identified [28]. Tiralongo et al. evaluated spontaneous retroperitoneal hematomas in 24 COVID-19 patients by a multiphasic CT and DSA. They report that multiphasic CT is a reliable diagnostic method and is applicable in 90% of cases. It is also beneficial to determine the anatomical extension and size of the hematoma to evaluate the effect of compression on the adjacent structures [27]. 

Still, there is no unified regiment of treatment for COVID-19-related spontaneous hematomas. The management of SRH is based on two established strategies–conservative non-invasive treatment and invasive operative treatment including interventional procedures and open surgery (Figure 4). In some cases, the successful treatments include administrations of vasopressors, blood product transfusions, fluids, and drugs that can reverse coagulopathy (vitamin K, protamine sulfate, prothrombin complex concentrates, recombinant factor VIII and IX). The rest of the patients are treated operatively depending on the volume of the hematoma and the subsequent abdominal hypertension [29].

## 4. Conclusions

The patient was admitted with real-time PCR showing positive results for SARS-CoV-2 infection, a generally worsening condition, biochemical parameters change, pneumonia, and no evidence of traumatic injury before hospitalization. During the hospital stay, the patient detected a drastic decrease in hemoglobin levels compared to the initial values accompanied by acute abdominal pain, which necessitated urgent diagnosis and subsequent surgery. The LMWH therapy in COVID-19 patients with pneumonia prevents thromboembolic events, but the risk of spontaneous bleeding should be not underestimated. Currently, an optimal anticoagulant dose in severe COVID-19 patients is not well established. The risk of spontaneous bleeding may be relevant or higher than the risk of thrombotic events, therefore is necessary to use an individual approach, careful clinical observation, and risk stratification of these patients. 

## Figures and Tables

**Figure 1 jcm-12-01842-f001:**
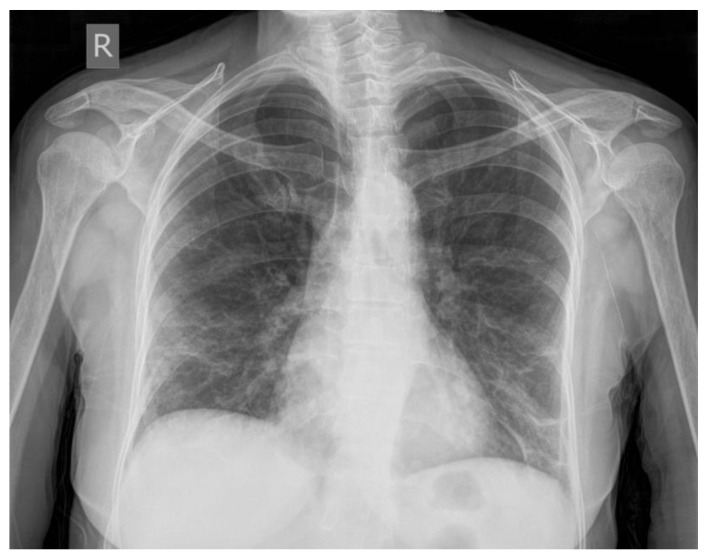
Chest X-ray examination of the bilateral COVID-19 pneumonia (on the 1st day of hospitalization). The X-ray images showed typical changes—“ground glass opacities” with consolidation at the bilateral lower lung area.

**Figure 2 jcm-12-01842-f002:**
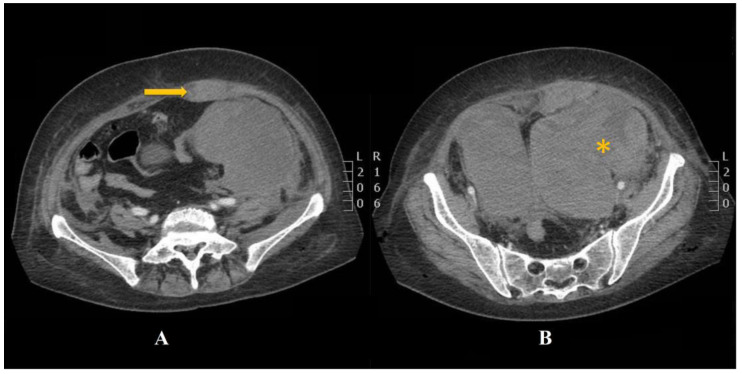
(**A**,**B**) Contrast-enhanced CT revealing a left rectus sheath hematoma (RSH); a thickened abdominal wall (**A**, yellow arrow); massive retroperitoneal bleeding extending down to the pelvis * (**B**); and imbibition of the left psoas muscle, internal oblique muscle. A homogeneous formation with approximate dimensions of 140 × 85 mm. Presence of free fluid along the intestinal loops and spleen on the left.

**Figure 3 jcm-12-01842-f003:**
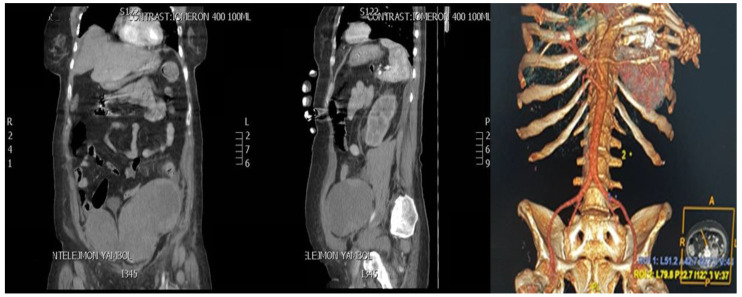
Contrast-enhanced CT of the abdominopelvic area: aorta and its branches. The CT images show a massive retroperitoneal hematoma dissecting down to the pelvis. Three-dimensional reconstruction indicates that the source of the bleeding does not originate from the main arterial abdominal vessels.

**Figure 4 jcm-12-01842-f004:**
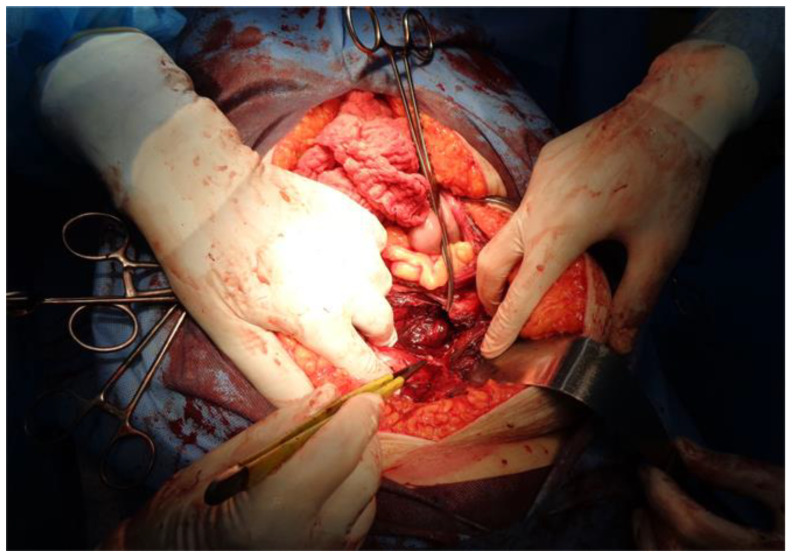
Intraoperative image of the left epigastric artery injury.

**Table 1 jcm-12-01842-t001:** Laboratory test results: 1st, 7th, and 10th hospitalization days. The biomedical data are shown as deviations from the reference ranges.

Test	1st Day	7th Day	10th Day	Unit	Reference Range
WBC	11.6	17.9	44.5	g/L	3.5–10.5
LYM	16	24	38.4	%	20–48
Neu	87.9	91.6	94.3	%	40–70
Mo	9.9	12.8	9.2	%	1–11
RBC	5.17	3.01	1.62	×10^12^/L	3.7–5.3
HGB	100	91	62	g/L	120–160
HCT	0.359	0.258	0.121	L/L	0.360–0.480
MCV	59.9	44.3	31.3	fL	80.0–96.0
MCH	26.5	28.3	28.3	pg	27.0–33.0
PLT	256	181	105	×10^9^/L	130–140
ESR	84	89	117	mm/h	0–20
Creatinine	290	390	849	µmol/l	65–127
Glucose	19.6	20.6	17.6	mmol/L	3.6–6.1
Urea	11.3	13.9	51.3	mmol/L	3.00–11.90
Uric acid	287	355	380	μmol/L	142–340
Total Bil	7.6	8.3	18.9	µmol/L	<21
TP	61.3	55.63	40.6	g/L	64.00–83.00
ASAT	21	31	177	U/L	<40.00
ALAT	23	139	323	U/L	<33.00
LDH	654	874	1152	U/L	<250
CK	65	83	766	U/L	<170.0
CRP	58.5	79.8	89.2	mg/L	<5
Na^+^	139	144	147	mmol/L	136.00–151.00
K^+^	3.8	4.9	6.1	mmol/L	3.50–5.60
D-dimer	1.54	2.08	2.47	mg/L	<0.5
Fib	3.6	2.4	2.52	g/L	2–4
INR	1.03	1,03	1.03	UI	0.8–1.2
PT	96	96	96	%	70–130
APTT	29.5	29.5	29.5	sec	27.6–37.2

## Data Availability

The personal data presented in this study are not publicly available due to privacy restrictions. The data are available on request from the corresponding author.

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
