# Peer review of "Case Report: Spontaneous Left Inferior Epigastric Artery Injury in a COVID-19 Female Patient Undergoing Anticoagulation Therapy"

_jcm, 2023, doi:10.3390/jcm12051842_

Round 1

Reviewer 1 Report

The work entitled "Case report: Spontaneous left inferior epigastric artery injury in  a COVID-19 female patient undergoing anticoagulation therapy" written in a clear and legible way. It presents the problem of potentially life-threatening bleeding complications of patients suffering from COVID 19, which are relatively rarely described. The paper lacks an clear explanation of the reason why the patient received a therapeutic dose of LMWH fraxiparine 2 x 0.6 ml at the beginning of hospitalization. Was it a standard procedure for patients with COVID-19 and multimorbidity ? The case report shows that the patient had no history of thromboembolic changes, she was not in intensive care at the time of admission to the hospital, and the level of d-dimer was elevated, but not significantly. In discussion section lacks information on the incidence of bleeding complications in the entire population of patients receiving anti-coagulant treatment and not necessarily suffering from COVID 19. What are the risk factors for these complications.

Author Response

Dear Reviewer,  

Thank you very much that you help us to improve our manuscript. All changes in the text are in blue color.

Reviewer 1

  1. The paper lacks an clear explanation of the reason why the patient received a therapeutic dose of LMWH fraxiparine 2 x 0.6 ml at the beginning of hospitalization. Was it a standard procedure for patients with COVID-19 and multimorbidity? The case report shows that the patient had no history of thromboembolic changes, she was not in intensive care at the time of admission to the hospital, and the level of d-dimer was elevated, but not significantly.
  2. 2. In discussion section lacks information on the incidence of bleeding complications in the entire population of patients receiving anticoagulant treatment and not necessarily suffering from COVID 19. What are the risk factors for these complications.

The answer to Point 1: D-dimer levels higher than 1.5 μg/ml may increase the incidence of venous thrombosis and a negative predictive value (94.7%) in severe COVID-19, accompanied by pneumonia. Hospitalized patients should receive a prophylactic dose of LMWH, except for those with active bleeding and a platelet count <25×109/l [11]. D-dimer levels in the patient we describe increased during the course of the disease despite the administered therapeutic dose of anticoagulant. Simultaneously, the PTL levels showed an increasing blood viscosity of 256x109/L (reference range 130 – 140x109/L) on the first day of hospitalization (Table 1). Based on the general worsening condition and multiple risk factors (pneumonia, elevated D-dimer 28.4% ≥2 times ULN, chronic diseases, and age > 60), the patient was assessed as at risk for thromboembolic events. This necessitated the introduction of a therapeutic anticoagulant regimen an according to recommendations for treating patients with moderate to severe coronavirus infection.

New references:

  1. Gómez-Mesa, J.E.; Galindo-Coral, S.; Montes, M.C.; Muñoz Martin, A.J. Thrombosis and Coagulopathy in COVID-19. Curr Probl Cardiol. 2021, 46, 100742. http://org.doi/10.1016/j.cpcardiol.2020.100742.
  2. Kollias, A.; Kyriakoulis, K.G.; Dimakakos, E;, Poulakou, G.; Stergiou, G.S.; Syrigos, K. Thromboembolic risk and anticoagulant therapy in COVID-19 patients: emerging evidence and call for action. Br J Haematol. 2020, 189, 846-847. http://org.doi/10.1111/bjh.16727.
  3. NIAID-RML, Credit. Coronavirus Disease 2019 (COVID-19) Treatment Guidelines. Nih. gov. Published, 2020.

Outside of the general manuscript: Anticoagulant therapy cannot be excluded due to the unpredictability of COVID-19 infection (including in some cases with low-risk patients). According to the inclusion and exclusion criteria defined in the Coronavirus Disease 2019 Treatment Guidelines, our patient meets all the criteria for the administration of a therapeutic anticoagulant dose, which is part of the standard therapy in patients with a middle and high risk of thrombotic events.

Answer of Point 2: The RSH risk factors include anticoagulation, concomitant therapy with fibrinolytic agents or glycoprotein inhibitors, increasing number of comorbidities, hematological disorders, surgical interventions, trauma, rectus muscle contractions, etc. [20, 21]. It is believed that the mortality rate in the general population is around 4% while in patients of anticoagulant therapy may increase to 25%. According to the literature, RSH is 2-3 times more frequently observed in women than men, and the mean patient's age varies from 46-69 years [22].

New references:

  1. Piran, S.; Schulman, S. Treatment of bleeding complications in patients on anticoagulant therapy. Blood. 2019, 133, 425-435. http://doi.org/10.1182/blood-2018-06-820746.
  2. Zandbaf, T.; Kalantari, M.E.; Azadmanesh, Y.; Sherafati, H.; Bagherzadeh, A. A. Heparin-induced rectus sheath hematoma in a COVID-19 patient with pulmonary emboli: a case report and literature review. Iranian Red Crescent Medical Journal. 2022, 24, e1727. https://doi.org/10.32592/ircmj.2022.24.4.1727.

Sincerely yours

Head Assist. Prof. Ekaterina Georgieva, Ph.D

Department of General and clinical pathology, forensic medicine, deontology and dermatovenerology, Faculty of Medicine, Trakia University, Stara Zagora, 6000 Bulgaria

Reviewer 2 Report

Thank you for giving me the opportunity to review this manuscript. The authors report a fatal case of a woman admitted with COVID-19 anticoagulation and developing serious retroperitoneal bleeding. Despite timely surgery and all efforts, the patient eventually died.

Comments:

The COVID-19 pandemic presented an unprecedented challenge to the medical community especially in the critical care field. Therefore, the aim of the authors to report an interesting case, which presented special challenge for their team is understandable. However, this case report has no novelty, there is nothing in it that has not been reported in earlier, much better designed studies. These are my further comments:

1.     Abstract is very poorly written and structured: the background is far too long, results do not allow any conclusions and the conclusion is basically missing.

2.     Materials and methods section is irrelevant in a case report

3.     Needs thorough English language editing

4.     ESR is rarely used these days especially not for infection diagnosis or monitoring inflammation

5.     Typo in line 112

6.     The conclusion contains general comments rather than drawing conclusion from this case. 

Author Response

Dear Reviewer, 

Thank you very much that you help us to improve our manuscript. All changes in the text are in red color.

Reviewer 2

  1. Abstract is very poorly written and structured: the background is far too long, results do not allow any conclusions and the conclusion is basically missing.

Answer 1: This part of the manuscript was shortened and revised according to the requirements of the reviewers.

  1. Materials and methods section is irrelevant in a case report

Answer 2: We exclude the Materials and methods section.

  1. Needs thorough English language editing

Answer 3: Done

  1. ESR is rarely used these days especially not for infection diagnosis or monitoring inflammation

Answer 4: We corrected

  1. Typo in line 112

Answer 5: Done

  1. The conclusion contains general comments rather than drawing conclusion from this case.

Answer 6:  We corrected the Conclusion

Conclusion: The patient was admitted with Real-time PCR showing positive results for COVID-19, a general worsening condition, changes in biochemical indicators, pneumonia, and no evidence of traumatic injury before hospitalization. During the hospital stay, was detect a drastic decrease in hemoglobin levels compared to the initial values accompanied by acute abdominal pain, which necessitated urgent diagnosis and subsequent surgery. The LMWH therapy in COVID-19 patients with pneumonia prevents thromboembolic still events risk of spontaneous bleeding should be not underestimated. Currently, an optimal anticoagulant dose in severe COVID-19 patients is not well established. The risk of spontaneous bleeding may be relevant or higher than the risk of thrombotic events, therefore is necessary to use an individual approach, careful clinical observation, and risk stratification of these patients.

Sincerely yours

Head Assist. Prof. Ekaterina Georgieva, Ph.D

Department of General and clinical pathology, forensic medicine, deontology and dermatovenerology, Faculty of Medicine, Trakia University, Stara Zagora, 6000 Bulgaria

Round 2

Reviewer 2 Report

The manuscript has certainly improved, and I don't think it can be improved further, apart from some more English editing (i.e.: line 38: "...which may initiate occur..." - the "initiate" should be deleted)